# 5-Methoxyindole, a Chemical Homolog of Melatonin, Adversely Affects the Phytopathogenic Fungus *Fusarium graminearum*

**DOI:** 10.3390/ijms222010991

**Published:** 2021-10-12

**Authors:** Mengmeng Kong, Jing Liang, Qurban Ali, Wen Wen, Huijun Wu, Xuewen Gao, Qin Gu

**Affiliations:** Department of Plant Pathology, College of Plant Protection, Nanjing Agricultural University, Key Laboratory of Monitoring and Management of Crop Diseases and Pest Insects, Ministry of Education, Nanjing 210095, China; 2017202025@njau.edu.cn (M.K.); 2020102011@stu.njau.edu.cn (J.L.); qurbanalirattar@webmail.hzau.edu.cn (Q.A.); 2019102015@njau.edu.cn (W.W.); hjwu@njau.edu.cn (H.W.); gaoxw@njau.edu.cn (X.G.)

**Keywords:** melatonin homolog, 5-methoxyindole, *Fusarium graminearum*, reactive oxygen species, cell death

## Abstract

*Fusarium graminearum* is a destructive fungal pathogen that threatens the production and quality of wheat, and controlling this pathogen is a significant challenge. As the cost-effective homolog of melatonin, 5-methoxyindole showed strong activity against *F. graminearum.* In the present study, our results showed the strong adverse activity of 5-methoxyindole against *F. graminearum* by inhibiting its growth, formation, and conidia germination. In addition, 5-methoxyindole could induce malformation, reactive oxygen species (ROS) accumulation, and cell death in *F. graminearum* hyphae and conidia. In response to 5-methoxyindole, *F. graminearum* genes involved in scavenging reactive oxygen species were significantly downregulated. Overall, these findings reveal the mechanism of antifungal action of melatonin-homolog 5-methoxyindole. To the best of our knowledge, this is the first report that a novel melatonin homolog confers strong antifungal activity against *F. graminearum*, and 5-methoxyindole is a potential compound for protecting wheat plants from *F. graminearum* infection.

## 1. Introduction

*Fusarium* head blight (FHB) caused by the filamentous fungus *Fusarium graminearum* Schwabe is a major disease on wheat (*Triticum aestivum* L.) and barley (*Hordeum vulgare*) and causes stalk and ear rot diseases in maize (*Zea mays*) and seedling blight in maize and wheat [1]. Several mycotoxins, such as deoxynivalenol (DON) and zearalenone (ZEN), produced by *F. graminearum* in infested grains can cause severe yield losses and grain quality reductions. Recently, disease severity of FHB has been reported to have increased globally in Asia, Canada, Europe, and South America. It has been identified as a major limiting factor in wheat production in many regions of the world [2]. Primary and secondary economic losses occurred due to FHB in all crops, especially in the Northern Great Plains and the Central United States (estimated to be USD 2.7 billion from 1998 to 2000) [3]. This disease has the capacity to destroy a potentially high-yielding crop within a few weeks of harvest, and these metabolites (DON and ZEN) represent a significant threat to the health of humans and animals, thus making the grain unhealthy for food or feed [4,5]. Large economic and health impacts caused by FHB and its management rely heavily on synthetic antifungal agents, such as benzimidazole fungicides, as no highly resistant varieties of wheat or barley are available [6,7,8,9]. However, the continuous use of these synthetic antifungal agents has resulted in the emergence of resistant *F. graminearum*, and poses a potential risk to the environment and human health [9,10,11]. Therefore, it is essential to develop alternative methods and molecular agents that have a low toxicity and are environmentally friendly, for the efficient control of FHB. 

Melatonin (N-acetyl-5-methoxytryptamine) is a ubiquitous molecule in animals and higher plants. In plant development, for example, it regulates seed germination, root development, photoprotection, flowering, leaf senescence, seed yield, and fruit ripening [12,13,14,15,16]. Plant melatonin also alleviates abiotic stress from cold, heat, drought, salt, and heavy metals [17,18,19,20]. Studies have also implicated the role of melatonin in plant immunity [21,22,23,24,25]. For example, phytomelatonin regulation of stomatal closure is dependent on its receptor CAND2/PMTR1-mediated H_2_O_2_ and the Ca^2+^ signaling transduction cascade after treatment with 10 µM melatonin [26], and 10 µM melatonin is responsible for rice resistance to rice stripe virus infection through a nitric oxide-dependent pathway [27]. Melatonin application reduces fluoride uptake and toxicity in rice seedlings by altering abscisic acid (ABA), gibberellin (GA), auxin, and antioxidant homeostasis after treatment with 20 and 100 µM melatonin, and its homologs induce immune responses via receptors trP47363-trP13076 in *Nicotiana benthamiana* [28,29].

Biotic stress is damage to crops caused by live organisms, such as fungi, bacteria, viruses, parasitic nematodes, insects, weeds and other indigenous plants [30]. Melatonin is considered a friendly molecule to the environment, and it plays an important role in plant protection against different biotic stressors. Melatonin has been reported to have antioxidant, anti-inflammatory, immunomodulatory, and neuroprotective activities in animals [31,32,33]. Many investigations have demonstrated the anti-pathogenic activity of melatonin in animals. For example, melatonin treatment could reduce the deleterious effects of reactive oxygen species (ROS) involved in the dissemination of the Venezuelan equine encephalomyelitis (VEE) virus, and melatonin administration also significantly decreased blood and brain viruses in comparison with infected control mice [31]. Melatonin also has defensive activities against bacterial infections in animals which have been tested under in vitro and in vivo conditions. The bactericidal capacity of melatonin was verified against multidrug-resistant Gram-negative and +positive bacteria, such as carbapenem-resistant *Pseudomonas aeruginosa* and *Acinetobacter baumannii*, as well as methicillin-resistant *Staphylococcus aureus* under in vitro conditions [34]. Furthermore, several studies support that melatonin plays a promotional role in the resistance of plants such as tomato (*Solanum lycopersicum*) fruit to *Botrytis cinerea* via the regulation of H_2_O_2_ generation and the jasmonic acid (JA) signaling mechanism [35]. Melatonin can increase fungicide susceptibility, enhance the vulnerability of *Phytophthora infestans* to different environmental stressors, decrease the dosage level, and promote the efficiency of fungicide treatment against potato late blight [36]. Similar results have been obtained in fungi, such as *Botrytis* spp, *Penicillium* spp, *Fusarium* spp, *Phytophthora nicotianae**,* and *Alternaria* spp [37,38,39]. The role of melatonin in the anti-pathogenic activity of viruses, bacteria, and fungi has been investigated, but only limited information is available on the activity of melatonin [31,34,37,38,39]. However, the direct inhibition mechanism of the melatonin homolog, 5-methoxyindole, has not been investigated in plant biotic stress, which could be a potential compound to protect wheat production from the stress of *F. graminearum* in the future.

In the present study, our results indicated that the melatonin homolog 5-methoxyindole shows stronger adverse activity to *F. graminearum* PH-1 compared with melatonin. The synthetic compound 5-methoxyindole also increased morphology malformation and accumulation of ROS in *F. graminearum* hyphae and conidia. Finally, 5-methoxyindole could directly induce apoptosis and cell death of *F. graminearum* hyphae and conidia. Collectively, the data suggest that melatonin homolog 5-methoxyindole has anti-phytopathogenic activities in the filamentous fungus *F. graminearum,* which may play an important role in controlling crop diseases.

## 2. Results and Discussion

### 2.1. 5-Methoxyindole Displayed Strong Antagonistic Activity against F. graminearum PH-1 Similar to Melatonin

Melatonin (N-acetyl-5methoxytryptamine), one of the best-studied biological molecules discovered from the pineal gland in the late 1950s, has been revealed to possess a broad spectrum of biological functions in both animals and plants in many studies [40,41,42,43,44]. In particular, MT has been considered a therapeutic indole for combating viral diseases, such as severe acute respiratory syndrome (SARS) and West Nile virus (WNV) [45], and has been shown to regulate circadian rhythms, the immune system, and induction of ROS, as well as sleep, food intake, mood, and body temperature in humans [46,47,48,49]. Consistent with the benefits of melatonin in immunology and medicine in animals, melatonin can also induce plant resistance to pathogens, such as *Pseudomonas syringae* DC3000, *Alternaria spp*, and *Fusarium spp* [39,50]. Melatonin has been reported to alleviate abiotic stress (cold, heat, drought, salt, and heavy metals) in several plants [17,18,19,20]. Twenty years after the discovery of melatonin in plants, the detection of melatonin in plant immunity has suggested specific physiological actions, where it can act as a growth regulator [41,51], improving plant growth parameters throughout plant development [12,13,14,15,16]. However, the direct inhibition mechanism of melatonin has not been investigated in plant biotic stress, and melatonin is expensive to produce, limiting its potential use in sustainable agriculture.

The impacts of several melatonin homologs on disease resistance against *P. nicotianae* and stomatal closure were investigated in *N*. *benthamiana*. The results showed that the melatonin homolog 5-methoxyindole could induce plant immunity by inducing stomatal closure, similar to melatonin in *N. benthamiana* [29]. Then, to identify the fungicidal activity of various concentrations of melatonin and its homolog 5-methoxyindole against *F. graminearum**,* water was used as a negative control and melatonin was used as a positive control. Antagonism assays showed that the melatonin homolog 5-methoxyindole significantly suppressed the growth of *F. graminearum* PH-1 in different media (PDA, CM and MM) similar to the melatonin (Figure 1A–D), the fungus could not grow at concentrations ≥2 mM 5-methoxyindole (Figure 1C). The results revealed that the melatonin homolog 5-methoxyindole had a more significant suppression ability against *F. graminearum* compared with melatonin at different concentrations (Figure 1E). In our initial results, 5-methoxyindole, as the cost-effective homolog of melatonin, showed strong activity against *F. graminearum* PH-1(Figure 1). These results were consistent with those reported for the bactericidal capacity of melatonin, which was verified against multidrug-resistant Gram-negative and -positive bacteria [34]. Until now, the mechanism of the activity of the melatonin homolog 5-methoxyindole has been poorly understood. Elucidation of the direct mechanism of the antifungal activity of melatonin homolog 5-methoxyindole is necessary for the efficient application of such compounds in crops.

### 2.2. Melatonin and 5-Methoxyindole Exhibited Strong Inhibitory Properties against the Conidiation and Germination of F. graminearum 

Melatonin is a common chemical in animals and higher plants, and numerous studies have shed light on its actions in vertebrate receptors. For example, two high-affinity G protein-coupled receptors (GPCRs) named MT1 (MT1a) and MT2 (MT1b) are involved in the perception of melatonin signaling in vertebrates [52,53]. Levoye et al. [54] identified GPR50, an orphan GPCR, as a member of the melatonin receptor subfamily. Additionally, a low-affinity melatonin binding site MT3 has been identified as the enzyme quinone reductase 2 [55]. However, compared with vertebrates, the function and signaling pathway of this putative phytohormone are largely undetermined due to the lack of identification of its receptor in plants. Recently, the first phytomelatonin receptor (CAND2/PMTR1) was identified in *A. thaliana* and was demonstrated to regulate stomatal closure induced by melatonin [26]. Melatonin has not been found to be involved in the receptors that work in plant pathogens, which is consistent with studies of melatonin in vertebrates and plants. As a result, the direct inhibition mechanism of melatonin and its cost-effective homolog 5-methoxyindole has potential in plants.

Conidial spore formation and germination represent the first steps triggering the asexual life cycle of *F. graminearum**,* and the formation and germination of conidial spores are important for *F. graminearum* spreading the disease in plants. In this study, both the formation and germination of conidial spores were inhibited strongly by 0.5 to 4 mM melatonin and 5-methoxyindole, respectively, compared with the control. In the control, mycelial growth formed normally, but the formation of *F. graminearum* mycelial growth and conidial spores was significantly inhibited after treatment with melatonin and 5-methoxyindole (Figure 2A,B), and conidial growth could not occur after treatment with 5-methoxyindole at concentrations ≥0.5 mM (Figure 2A). Furthermore, the germination of *F. graminearum* conidia was inhibited by melatonin and 5-methoxyindole in a dose-dependent manner. With 1 mM melatonin and 5-methoxyindole, the germination rate was 27.32 and 4.46%, respectively (Figure 2B), which showed that the melatonin, 5-methoxyindole, has stronger activity against the germination of conidial spores in *F. graminearum* (Figure 2B). In addition, because of the stronger inhibitory properties of 5-methoxyindole on the *F.**graminearum* hyphae and conidial spores compared with melatonin, 5-methoxyindole, a cost-effective molecular agents of melatonin, should be considered for potential use in the plants to inhibit fungal diseases. 

### 2.3. Melatonin and 5-Methoxyindole Released the Inhibitory Function of H_2_O_2_ against F. graminearum

H_2_O_2_, Calcofluor white stain (CFW), and Congo red are fungal compound inhibitors of *F. graminearum* PH-1. To analyze the function of melatonin and 5-methoxyindole, they were compared with the compound inhibitors (H_2_O_2_, CFW and Congo red) in *F. graminearum*. *F. graminearum* was grown in pretreated together with the melatonin, 5-methoxyindole and the fungus compound inhibitors (H_2_O_2_, CFW and Congo red) for 3 days. The results showed that melatonin could release the inhibitive ability of H_2_O_2_, and more significantly enhance the growth of *F. graminearum* after the combined treatment of Congo red and melatonin or Congo red and 5-methoxyindole. However, it had no effect on CFW inhibitor ability when treated with melatonin and CFW together (Figure 3A,B). 5-methoxyindole released the suppressive ability of H_2_O_2_, similar to melatonin at a low concentration and enhanced the suppressive ability of *F. graminearum* after treatment with Congo red and 5-methoxyindole together followed by CFW and 5-methoxyindole together (Figure 3B).

### 2.4. Melatonin and 5-Methoxyindole Led to Morphologic Changes in F. graminearum Hyphae and Conidia

In the present results, the formation of *F. graminearum* conidial spores was inhibited significantly with 4 mM melatonin and 1 mM 5-methoxyindole, but conidia production was decreased largely after being treated with concentrations of ≥1 mM melatonin. In addition, the conidia could not be produced after treatment with 5-methoxyindole at concentrations ≥0.5 mM (Figure 2). Then, the effect of *F. graminearum* hyphae and conidia was measured after being treated with 4 mM melatonin/1 mM 5-methoxyindole and 1 mM melatonin/0.5 mM 5-methoxyindole, respectively. Various studies have shown that melatonin plays an important role in plant resistance to pathogens by promoting plant immunity [35], and melatonin can also increase fungicide susceptibility, enhance the vulnerability of plant-pathogens to different environmental stressors, and decrease the dosage level [36]. The 5-methoxyindole, as a cost-effective homolog of melatonin, demonstrated strong activity against *F. graminearum* conidiation and germination in our preliminary findings (Figure 2A,B). Electron microscopy was employed to better understand the mechanism of morphological changes in *F. graminearum* hyphae and conidia and the direct mechanism of melatonin and 5-methoxyindole against *F. graminearum*.

Our results showed that clear morphological variations of fungal mycelia and spores were observed by microscopic observations treated with 4 mM melatonin and a relatively low concentration of 1 mM 5-methoxyindole. The structure of untreated *F. graminearum* mycelia was well organized, while structures appeared swollen in the presence of melatonin and 5-methoxyindole (Figure 4A). These swollen structures were observed both at the tip and in the central part of the treated hyphae. A similar phenomenon was also found in the conidia (Figure 4B). The untreated conidia germinated and appeared normal. After treatment with 1 mM melatonin and 0.1 mM 5-methoxyindole, conidia germination was inhibited, and the conidia that were produced appeared swollen (Figure 4B). In addition, the malformation rate of swollen conidia was significantly higher than the control after being treated with melatonin and 5-methoxyindole (Figure 4C,D). These observations also indicate that melatonin and 5-methoxyindole could cause severe damage to *F. graminearum* hyphae and conidia, thus inhibiting the phytopathogenic fungus. 

### 2.5. Accumulation of Reactive Oxygen Species by F. graminearum Hyphae and Conidia Treated with Melatonin and 5-Methoxyindole

ROS (e.g., O^2.−^, H_2_O_2_, OH^˙^, 1 O_2_) are partially reduced or activated forms of atmospheric oxygen (O_2_) [56]. When ROS concentrations increase in cells, they cause oxidative damage to membranes (lipid peroxidation), proteins, RNA and DNA molecules, and can even cause the oxidative destruction of the cell in a process termed oxidative stress, such as drought, heat, salinity, and high light [57]. ROS play a key role in the acclimation process of plants to abiotic stress. They primarily function as signal transduction molecules which regulate different pathways during plant acclimation to stress, but are also toxic byproducts of stress metabolism [58]. Low ROS concentrations act as intracellular messengers for many molecular events; however, large amounts of ROS are associated with cell death [59].

To investigate whether *F. graminearum* cells accumulate ROS as a result of melatonin and 5-methoxyindole treatment, an ROS assay kit was used. The treated hyphae and conidia of *F. graminearum* appeared to emit stronger green fluorescence, compared with the untreated control. In particular, hyphae and conidia of *F. graminearum* showed green fluorescence (Figure 5A,B). The average fluorescence of the hyphae and conidia was also stronger than that of the control after measurement of the unit area of fluorescence when the fungus was treated with melatonin and 5-methoxyindole, respectively (Figure 5C–F). In conclusion, the results showed that 5-methoxyindole largely induced ROS accumulation in the hyphae and conidia of *F. graminearum*, similar to melatonin. Further study of the mechanism of action of melatonin and its homolog, 5-methoxyindole, showed that both of the compounds induced high ROS accumulation, which is associated with cell death in *F. graminearum* hyphae and conidia.

### 2.6. Extracellular ROS-Scavenging Enzyme Genes and Peroxidase Genes of F. graminearum Were Downregulated in the Presence of Melatonin and 5-Methoxyindole

*F. graminearum* contains five putative extracellular ROS-scavenging enzymes, i.e., three putative catalases (FGSG_02881, FGSG_06554, and FGSG_06733) and two putative peroxidases (FGSG_02974 and FGSG_12369) [60]. The results of quantitative real-time PCR (qRT-PCR) analysis showed that all five genes in *F. graminearum* were significantly downregulated after treatment with melatonin and 5-methoxyindole, especially those for two of the catalases (FGSG_12369 and FGSG_02881) (Figure 6A,B). Further study of the mechanism of melatonin and its homolog, 5-methoxyindole, showed that both of two compounds induced high ROS accumulation in *F. graminearum* hyphae and conidia (Figure 5). Additionally, melatonin and its homolog, 5-methoxyindole, induced the expression of the glutathione reductase and thioredoxin genes in *F. graminearum*, which are involved in ROS synthesis.

### 2.7. Apoptosis of F. graminearum Hyphae and Conidia Was Caused by Melatonin and 5-Methoxyindole

Investigations have demonstrated that large amounts of ROS are associated with cell death [59]. The induction of ROS bursts by iturins has been suggested to contribute to cell death in *V. dahlia* [61]. In our results, both melatonin and its homolog 5-methoxyindole induced high ROS accumulation, which is associated with cell death in *F. graminearum* hyphae and conidia (Figure 5 and Figure 6). To further study the mechanism, which has strong activity against *F. graminearum* caused by melatonin and its homolog 5-methoxyindole, Hoechst fluorescence staining was used to analyze apoptosis of *F. graminearum* in combination with phase-contrast and fluorescence microscopy. Hoechst is a compound that releases blue fluorescence once the compound enters living cells, and the compound thus serves as an indicator of apoptotic cells. The results showed that apoptosis not only occurred in *F. graminearum* hyphae after being treated with melatonin and 5-methoxyindole (Figure 7A), but also occurred in *F. graminearum* conidia (Figure 7B). After measuring the unit area of fluorescence in *F. graminearum* hyphae and conidia, melatonin and 5-methoxyindole induced fluorescence accumulation in *F. graminearum* hyphae and conidia (Figure 7C–F). 

### 2.8. Melatonin and 5-Methoxyindole Induced the Cell Death of F. graminearum Hyphae and Conidia

Many studies have demonstrated that low ROS concentrations act as signaling messengers for many molecular events in plants; however, large amounts of ROS could induce cell death [59]. Propidium iodide (PI) emits red fluorescence in response to membrane damage and is used as an indicator of dead cells. As shown in Figure 8, the untreated hyphae and conidia had few dead cells (red fluorescence). Accordingly, their typical shape, outlined by blue fluorescence (apoptotic cells), could be identified easily (Figure 7). In contrast, after 3 days of exposure to melatonin and 5-methoxyindole, the proportion of cells with red fluorescence in the hyphae and conidia increased (Figure 8A,B). After measuring the unit area of fluorescence in *F. graminearum* hyphae and conidia, melatonin and 5-methoxyindole induced red fluorescence accumulation in *F. graminearum* hyphae and conidia (Figure 8C–F). As shown in Figure 7 and Figure 8, the untreated hyphae and conidia were rarely apoptotic (blue fluorescence) and had few dead cells (red fluorescence). A large amount of blue and red fluorescence accumulated in the hyphae and conidia of *F. graminearum* after treatment with melatonin and its homolog 5-methoxyindole. These all the results demonstrated that 5-methoxyindole, a cost-effective homolog of melatonin, has strong activity against the fungus *F. graminearum**,* inducing apoptosis and killing cells in hyphae and conidia (Figure 7 and Figure 8). Collectively, our data indicate that mechanism of strong antifungal activity against *F. graminearum* by melatonin and 5-methoxyindole through inhibiting properties against conidiation and germination, inducing ROS accumulation, apoptosis, and cell death in *F. graminearum* hyphae and conidia. In addition, this is the first report that a novel melatonin homolog confers strong antifungal activity against *F. graminearum*, and 5-methoxyindole is a potential compound for protecting wheat plants from *F. graminearum* infection.

## 3. Materials and Methods

### 3.1. Fungal Strains and Growth Conditions

For conidial spore cultures, fresh mycelia of each strain (50 mg), taken from the periphery of a 3-day-old colony, were inoculated in a 50 mL flask containing 20 mL mung bean liquid (MBL) broth (10 g of mung beans were boiled in 1 liter of water for 20 min and filtered through cheesecloth). After incubation at 25 °C for 4 days in a shaker (180 rpm), the number of conidia in each flask was determined using a hemocytometer.

### 3.2. Antifungal Activity Assay and EC50 Determination

The EC50 of melatonin and 5-methoxyindole were evaluated as follows. A 0.6 cm-diameter plug containing mycelium was placed at the center of potato dextrose agar (PDA) plates containing different concentrations of melatonin and 5-methoxyindole (0, 0.1, 0.5, 1, 2, and 4 mM), and the plates were incubated at 25 °C for 48 h. The diameter of the colony was measured, and the EC50 was calculated using statistical analysis the error bars represent the mean standard deviation of each treatment, which was repeated three times with three replicates. The letters above the columns indicate significant differences. The significant difference between the treatments was determined through Tukey’s HSD test at *p* ≤ 0.05 [62]. 

### 3.3. Fungus Inhibitor Compound Assays with Melatonin and 5-Methoxyindole against F. graminearum

A 0.6 cm-diameter plug containing mycelium of *F. graminearum* was placed at the center of CM (complete medium) plates containing different concentrations of melatonin, 5-methoxyindole (0, 0.1, 0.5, 1, 2, and 4 mM), and fungus compound inhibitors (H_2_O_2_, CFW, and Congo red) together, and the plates were incubated at 25 °C for 3 days. The diameters of the fungal colonies were measured and the EC50 was calculated using statistical analysis [62]. The experiment was repeated three times. 

### 3.4. Laser Scanning Confocal Microscopic Observation of Hyphal and Conidial Morphologies

To observe the morphological changes in hyphae and conidia caused by melatonin and 5-methoxyindole, laser scanning confocal microscopy (LSM) was used. For laser scanning confocal microscopy, hyphae and conidia were treated with different concentrations of melatonin and 5-methoxyindole. Three days after treatment, the hyphae and conidia were observed using an LSM 880 with an Airyscan microscope. 

### 3.5. Reactive Oxygen Species Detection

For the detection of reactive oxygen species (ROS) assay kit was used in the cell of *F. graminearum* hyphae and conidia accumulated after treatment with melatonin and 5-methoxyindole, the probe dichlorodihydrofluorescein diacetate (DCFH-DA) and fluorescence microscopy were used. The *F. graminearum* hyphae and conidia were treated with melatonin and 5-methoxyindole for 5 h, centrifuged at 1000× *g* for 10 min, and resuspended in 10 mM sodium phosphate buffer (pH 7.4). The samples were then incubated with 10 µM DCFH-DA for 30 min at 19 to 21 °C [63]. The samples were viewed with an LSM 880 with Airyscan microscope (excitation, 488 nm; emission, 535 nm).

### 3.6. Formation and Germination of Conidia

To elevate the formation of the conidia, CMC medium (15 g sodium carboxymethyl cellulose, 1 g yeast extract, 1 g NH_4_NO_3_, 1 g KH_2_PO_4_, 0.5 g MgSO_4_·7H_2_O, and 1 liter water) was used [62]. Five fresh mycelial plugs of *F. graminearum*, taken from the periphery of a 3-day-old colony, were inoculated into 50-mL flasks containing 20 mL of CMC medium with 0.05% (*v*/*v*) methanol/water and different concentrations of melatonin and 5-methoxyindole (0, 0.1, 0.5, 1, 2, and 4 mM). Twenty milliliters of CMC medium with 0.05 (*v*/*v*) methanol/water served as the control. After incubation at 25 °C for 3 days in a shaker (180 rpm), the number of conidia in each flask was determined using a hemocytometer. Three repeats were performed. For the assay of conidial spore germination, 1 mL of conidial suspension (10^3^ conidia/mL) containing 0.05% (*v*/*v*) methanol/water and different amounts of melatonin and 5-methoxyindole (0, 0.1, 0.5, 1, 2 and 4 mM) were incubated at 25 °C for 24 h in a shaker (180 rpm) before counting. One milliliter of conidial suspension (10^3^ conidia/mL) with 0.05% (*v*/*v*) methanol/water served as the control. The experiment was repeated three times.

### 3.7. Live/Dead Fungus Viability Staining

The cell viability assay was performed using a blue fluorescein Hoechst stain and a red fluorescent propidium iodide stain [64]. When the stains are used in an appropriate mixture, live fungal cells with intact membranes show blue fluorescence, while fungal cells with damaged membranes show red fluorescence. The hyphae and conidia of *F. graminearum* that had been treated with melatonin and 5-methoxyindole for 3 days were centrifuged at 1000× *g* for 10 min and resuspended in 10 mM sodium phosphate buffer (pH 7.4). Then, 10 µL of the fluorescein Hoechst and propidium iodide molecular probes, prepared as recommended by the manufacturer, were added, and the cell suspensions were incubated for 15 min at 25 °C in the dark. The samples were viewed using an LSM 880 with Airyscan microscope.

### 3.8. RNA Isolation and Reverse Transcription-Quantitative PCR (RT-qPCR)

To extract total RNA, the mycelia of *F. graminearum* PH-1 were inoculated in potato dextrose broth (PDB) and cultured for 2 days at 25 °C in the dark. Melatonin and 5-methoxyindole were then added to the flask for another 2 h, with methanol serving as the control. Mycelia were harvested by filtration using two layers of Miracloth and were washed with sterilized water. Harvested mycelia were then lyophilized and ground in liquid nitrogen. Total RNA was extracted from the mycelia using the TaKaRa RNAiso reagent (TaKaRa Biotechnology Co., Dalian, China), according to the manufacturer’s instructions. First-strand cDNA was synthesized using reverse transcriptase (TaKaRa) with random hexamer primers. The resulting cDNA was used as the template for subsequent PCR amplification. qRT-PCR was performed using SYBR Premix ExTaq (TaKaRa) in a 7500 fast real-time PCR detection system. Primers were designed through the Primer Quest tool for selected genes listed in Appendix A (see Appendix A) [60]. The housekeeping gene actin previously described in *F. graminearum* as endogenous control was used in the present study [62]. Finally, relative quantification was performed based on the comparative C method of 2^−ΔΔCT^ as described by [65].

### 3.9. Statistical Analysis

All experiments were conducted in a completely randomized design. Experimental data were subjected to statistical analysis using Statistics 8.1 (analytical software, USA) [66]. Means were separated using Tukey’s HSD test at *p* ≤ 0.05 after ANOVA.

## 4. Conclusions

The 5-methoxyindole, the basic functional homolog of the melatonin, showed a strong adverse activity to *F. graminearum* PH-1 followed by melatonin. Both melatonin and 5-methoxyindole had strong activity against *F. graminearum* through inhibiting the fungus growth, formation and germination of *F. graminearum* conidia and hyphae. In addition, melatonin and 5-methoxyindole could also induce the malformation, accumulation of ROS and caused cell death in *F. graminearum* hyphae and conidia. Overall, the findings imply that the melatonin homolog, 5-methoxyindole, has anti-phytopathogenic activities in the filamentous fungus *F. graminearum* which could be useful in disease control in sustainable agriculture.

## Figures and Tables

**Figure 1 ijms-22-10991-f001:**
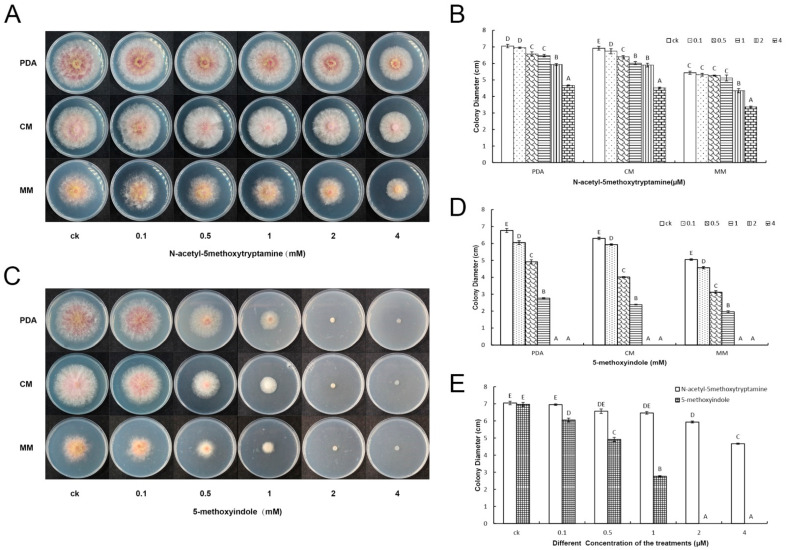
The antagonistic activity of melatonin and the 5-methoxyindole toward *F. graminearum* PH-1. (**A**,**B**) *F. graminearum* has grown in different media (PDA, CM and MM) containing different concentrations of melatonin (0, 0.1, 0.5, 1, 2 and 4 mM) grown for 3 days, and EC50 of melatonin were evaluated. (**C**,**D**) *F. graminearum* has grown in different medias containing different concentrations of 5-methoxyindole grown for 3 days, and the EC50 of 5-methoxyindole were evaluated. (**E**) The EC50 of melatonin and 5-methoxyindole were evaluated in the media of PDA. The experiment was repeated independently three times, and 0.05% (vol/vol) methanol/water served as the ck. The error bars represent the mean standard deviation of each treatment repeated three times with three replicates. The letters above the columns indicate significant differences. The significant difference between the treatments was determined through Tukey’s HSD test at *p* ≤ 0.05.

**Figure 2 ijms-22-10991-f002:**
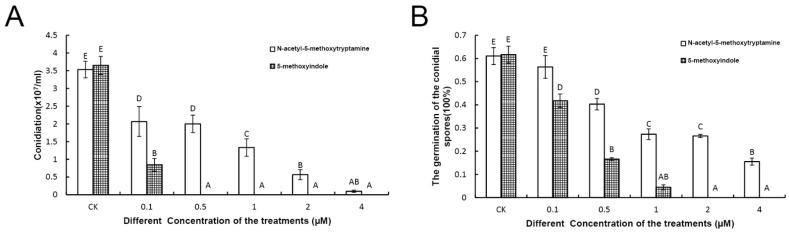
Analysis of *F. graminearum* PH-1 conidiation treated with different concentrations of melatonin and 5-methoxyindole (**A**). The conidial spore germination of *F. graminearum* after being treated with the different concentrations of melatonin and the 5-methoxyindole (**B**). The experiment was repeated independently three times, and 0.05% (vol/vol) methanol/water served as the ck. The error bars represent the mean standard deviation of each treatment repeated three times with three replicates. The letters above the columns indicate significant differences. The significant difference between the treatments was determined through Tukey’s HSD test at *p* ≤ 0.05.

**Figure 3 ijms-22-10991-f003:**
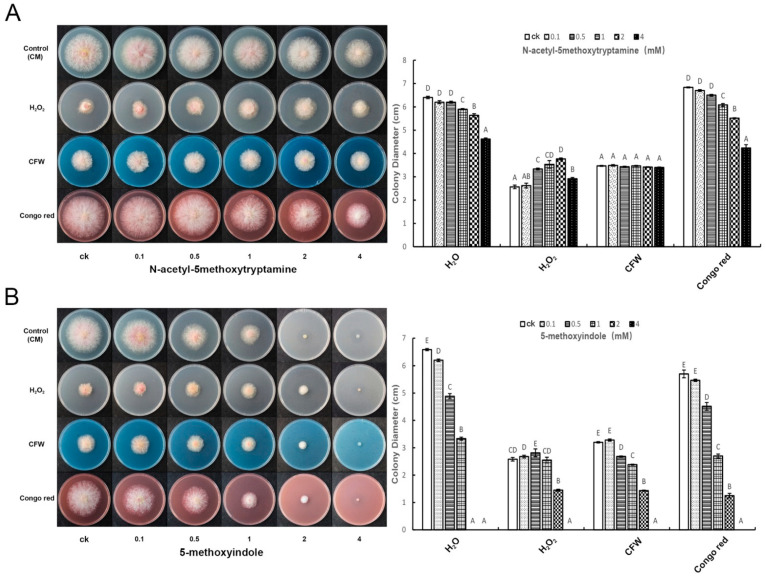
The functional analysis of pretreated melatonin and 5-methoxyindole with the fungus compound inhibitors (H_2_O_2_, CFW and Congo red) together. (**A**) The fungus *F. graminearum* PH-1 was grown in the medium CM after being treated with using melatonin and fungus compound inhibitors (H_2_O_2_, CFW, and Congo red) together for 3 days. (**B**) The fungus *F. graminearum* growth in the medium CM after treating 5-methoxyindole with the fungus compound inhibitors (H_2_O_2_, CFW, and Congo red) together for 3 days. The experiment was repeated independently three times, and 0.05% (vol/vol) methanol/water served as the ck. The error bars represent the mean standard deviation of each treatment repeated three times with three replicates. The letters above the columns indicate significant differences. The significant difference between the treatments was determined through Tukey’s HSD test at *p* ≤ 0.05.

**Figure 4 ijms-22-10991-f004:**
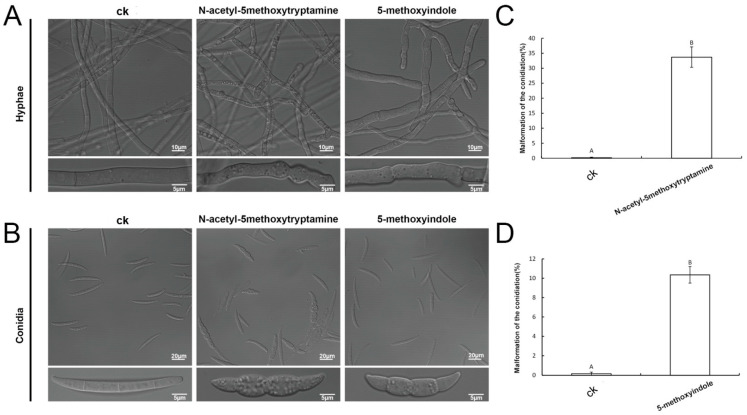
Effect of the melatonin and the 5-methoxyindole on the morphology of *F. graminearum* PH-1 hyphae and the conidia. (**A**) Effect of 4 mM melatonin and the 1 mM 5-methoxyindole on the morphology of *F. graminearum* hyphae observed with a microscope of the LSM 880 with Airyscan. (**B**)The morphological changes of *F. graminearum* conidia treated with the 1 mM melatonin and the 0.1 mM 5-methoxyindole grown for 3 days. (**C**,**D**) The swollen rate of conidia after treatment with melatonin and 5-methoxyindole grown for 3 days. The experiment was repeated independently three times, and 0.05% (vol/vol) methanol/water served as the ck. The error bars represent the mean standard deviation of each treatment repeated three times with three replicates. The letters above the columns indicate significant differences. The significant difference between the treatments was determined through Tukey’s HSD test at *p* ≤ 0.05.

**Figure 5 ijms-22-10991-f005:**
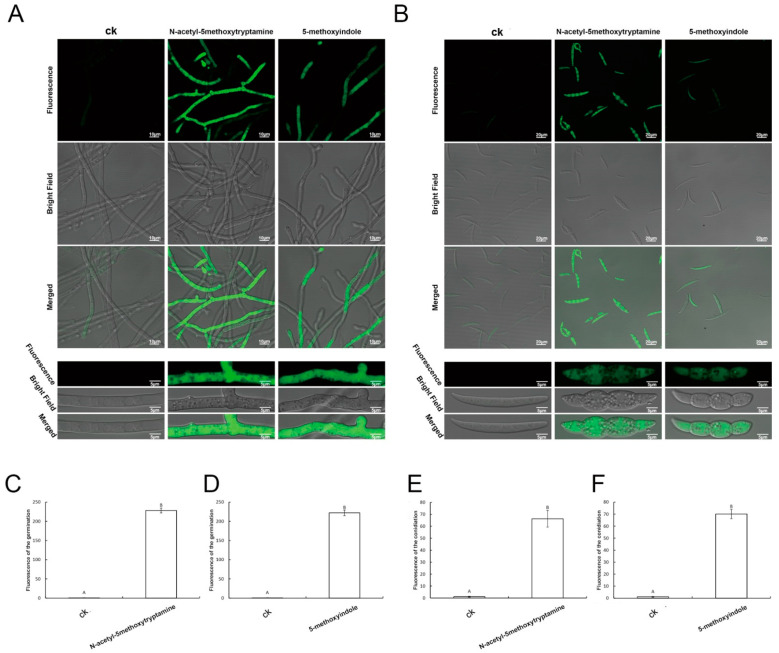
The ROS production of *F. graminearum* PH-1 hyphae and conidia after being treated with melatonin and 5-methoxyindole. (**A**,**B**) The ROS detection of *F. graminearum* hyphae and conidia were based on DCFH-DA staining after treatment with 4 mM melatonin and the 0.1 mM 5-methoxyindole grown for 3 days; (**C**,**D**) Average fluorescence analysis of *F. graminearum* hyphae after treatment with the melatonin and 5-methoxyindole grown for 3 days; (**E**,**F**) average fluorescence analysis of *F. graminearum* conidia after treatment with the melatonin and 5-methoxyindole grown for 3 days. The experiment was repeated independently three times, and 0.05% (vol/vol) methanol/water served as the ck. The error bars represent the mean standard deviation of each treatment repeated three times with three replicates. The letters above the columns indicate significant difference. The significant difference between the treatments was determined through Tukey’s HSD test at *p* ≤ 0.05. The higher level of green fluorescence indicates the high level of ROS induction in *F. graminearum* exposed to melatonin and 5-methoxyindole.

**Figure 6 ijms-22-10991-f006:**
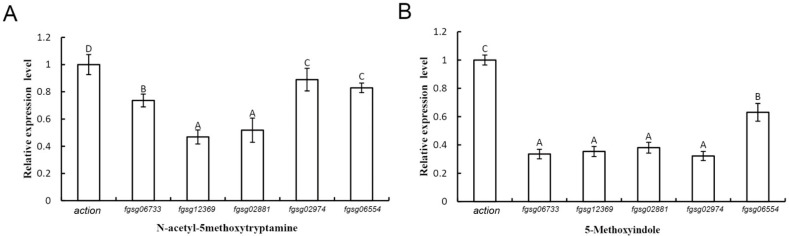
Quantitative real-time PCR analysis of the expression of five genes (*fgsg_02881, fgsg_02974, fgsg_06554, fgsg_06733, and fgsg_12369*) in *F. graminearum* PH-1 in response to 4 mM melatonin (**A**) and the 0.1 mM 5-methoxyindole (**B**) treatment. Values were normalized to the levels of the actin gene as an internal reference. The y axis represents the mean expression values standard deviations (SDs), relative to the control. The experiments were repeated independently three times. In all of the experiments, 0.05% (vol/vol) methanol/water served as the ck. The letters above the columns indicate significant differences.

**Figure 7 ijms-22-10991-f007:**
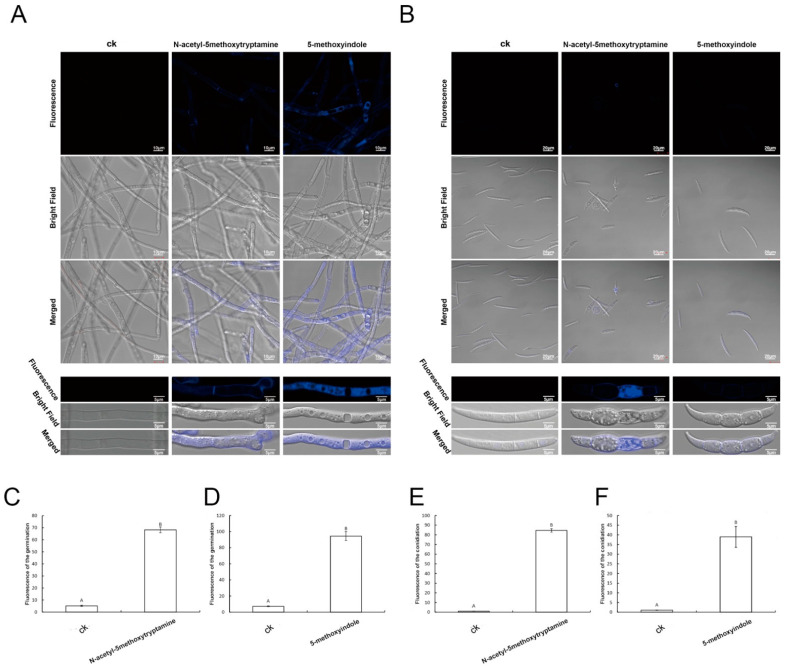
(**A**,**B**) Detection of *F. graminearum* hyphae and conidia viability based on fluorescein Hoechst staining after treatment with 4 mM melatonin and 0.1 mM 5-methoxyindole for 3 days; apoptotic cells with intact membranes show blue fluorescence; 0.05% (vol/vol) methanol/water served as the ck; (**C**,**D**) analysis of the average fluorescence of the *F. graminearum* hyphae after treating with the melatonin and 5-methoxyindole for 3 days; (**E**,**F**) analysis of the average fluorescence of the *F. graminearum* conidia after treating with the melatonin and 5-methoxyindole for 3 days. The error bars represent the mean standard deviation of each treatment repeated three times with three replicates. The letters above the columns indicate significant differences. The significant difference between the treatments was determined through Tukey’s HSD test at *p* ≤ 0.05. The higher level of green fluorescence indicates the high level of ROS induction in *F. graminearum* exposed to the melatonin and 5-methoxyindole.

**Figure 8 ijms-22-10991-f008:**
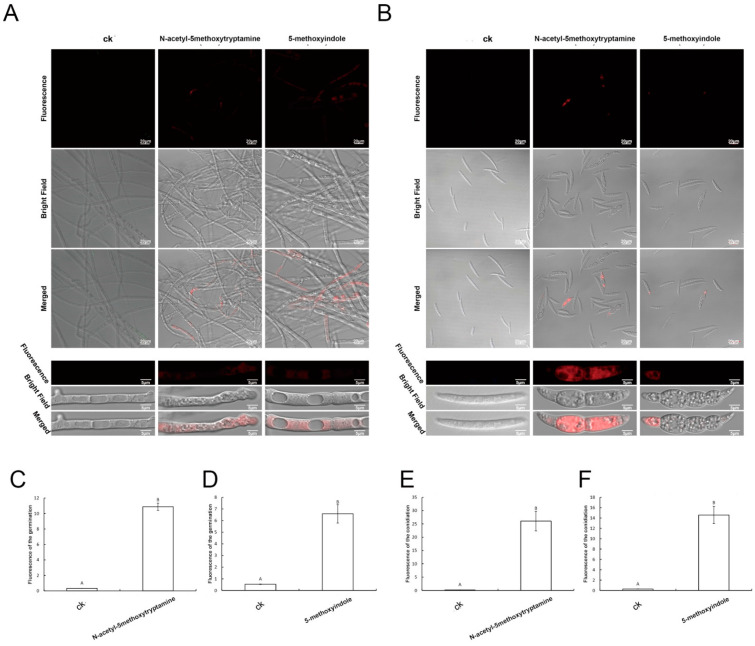
(**A**,**B**) Detection of *F. graminearum* hyphae and conidia viability based on propidium iodide staining after treatment with 4 mM melatonin and the 0.1 mM 5-methoxyindole for 3 days. Fungal cells with damaged membranes show red fluorescence; 0.05% (vol/vol) methanol/water served as the ck; (**C**,**D**) Analysis of the average fluorescence of the *F. graminearum* hyphae after treating with the melatonin and 5-methoxyindole for 3 days; (**E**,**F**) Analysis of the average fluorescence of the *F. graminearum* conidia after treating with the melatonin and 5-methoxyindole for 3 days. The error bars represent the mean standard deviation of each treatment repeated three times with three replicates. The letters above the columns indicate significant differences. The significant difference between the treatments was determined through Tukey’s HSD test at *p* ≤ 0.05. The higher level of green fluorescence indicates the high level of ROS induction in *F. graminearum* exposed to the melatonin and 5-methoxyindole.

## Data Availability

Not applicable.

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
