# Peer review of "5-Methoxyindole, a Chemical Homolog of Melatonin, Adversely Affects the Phytopathogenic Fungus Fusarium graminearum"

_ijms, 2021, doi:10.3390/ijms222010991_

Round 1
Reviewer 1 Report
This ms can be published in its current status.
Author Response
Reviewer 1:
Point 1
This MS can be published in its current status.
Response: Thank you very much for your review of our manuscript and we are very appreciating to get you approve of our submitted manuscript.

Reviewer 2 Report
Dear Authors,
The article submitted for review is indeed the first work on the use of 5-methoxyindole to combat Fusarium graminearum in the environment to protect agricultural crops. Thus, the work presented here is an interesting discussion of two issues: effective fungus control and the use of environmentally friendly agents.
My comments:
1. Introduction part
A). The authors state that Fusarium graminearum is a major problem in agriculture but do not state the extent of this threat to crops. I, therefore, ask you to correct this part of the literature introduction with data.. This will allow better visualization of the problem.
B). My curiosity was piqued by the description of melatonin and its uses while the manuscript deals with 5-methoxyindole. Can the authors similarly describe the use of the chemical compound they use in their study to get rid of the fungus?
2. Materials and Methods part
Well organized part. However, Sections 4.2 and 4.3 as well as 4.6 need references.
3. Results and discussion part
This part very thoroughly describes the measurement results obtained. Each section is followed by a good commentary and discussion.
A). However, since the authors claim that their results reveal a mechanism for the antifungal action of the melatonin-homologue 5-methoxyindole, I would request that a graphical version of this mechanism be prepared and included in the manuscript.
Authors, please improve the manuscript in the major mode.
Author Response
Reviewer 2:
Point 1
Introduction part A). The authors state that Fusarium graminearum is a major problem in agriculture but do not state the extent of this threat to crops. I, therefore, ask you to correct this part of the literature introduction with data. This will allow better visualization of the problem.
Response: Thank you very much for your helpful suggestions and valuable input in our research manuscript. We have modified and corrected this part of the literature introduction with data in our manuscript, see the lines (30-34)
Point 2
B). My curiosity was piqued by the description of melatonin and its uses while the manuscript deals with 5-methoxyindole. Can the authors similarly describe the use of the chemical compound they use in their study to get rid of the fungus?
Response:Thanks, in our present study, our results indicated melatonin homolog, 5-methoxyindole, shows the stronger adverse activity to F. graminearum PH-1 as compared with melatonin. The synthetic compound 5-methoxyindole also increased morphology malformation and accumulation of ROS of F. graminearum hyphae and conidia as same as melatonin. Finally, we also identified 5-methoxyindole could directly induce the apoptosis, cell death of F. graminearum hyphae and conidia which were similar with melatonin. Collectively, the data suggest that melatonin homolog, 5-methoxyindole, have anti-phytopathogenic activities in the filamentous fungus F. graminearum, which may take the place role in controlling crop diseases. see the lines (88-95).
Point 3
- Materials and Methods part
Well organized part. However, Sections 4.2 and 4.3 as well as 4.6 need references.
Response:Thanks, we have added the references in the Sections 4.2 and 4.3 as well as 4.6 of our manuscript, see the lines (388, 398, 418)
Point 4
- Results and discussion part
This part very thoroughly describes the measurement results obtained. Each section is followed by a good commentary and discussion. However, since the authors claim that their results reveal a mechanism for the antifungal action of the melatonin-homologue 5-methoxyindole, I would request that a graphical version of this mechanism be prepared and included in the manuscript. Authors, please improve the manuscript in the major mode.
Response:Thanks, we have made a graphical version of antifungal action of the melatonin-homologue 5-methoxyindole mechanism in our research manuscript (Graphical Abstract, see Supporting Information).
Reviewer 3 Report
Please read the manuscript and the recommended comments and changes. Other comments Page 5Fig 2 what it mean "ck" is it control?
This manuscript in my opinion has many short comes. First of all they used "malatonin" as positive control, and they used it's chemical homolog 5-methoxyindol as chemical against fusarium head blight. In Positive control you use a registered synthetic chemical that is proven to control fusarium head blight for comparative control with the tested material. fusarium head blight is only tested in laboratory only on artificial medium and not on the plants e.g. wheat or barley. I cannot provide any comments on statistical analyses, you should seek the help of an statistician. As negative control they used only pure water in one stage in another stage authors mention methanol mixed with water.
Author Response
Reviewer 3:
Point 1
Please read the manuscript and the recommended comments and changes. Other comments Page 5Fig 2 what it means "ck" is it control?
Response:Thank you very much for your helpful suggestions, we have modified the “control” to “ck” in all the figures of our whole manuscript.
Point 2
This manuscript in my opinion has many short comes. First of all they used "malatonin" as positive control, and they used it's chemical homolog 5-methoxyindol as chemical against Fusarium head blight. In Positive control you use a registered synthetic chemical that is proven to control Fusarium head blight for comparative control with the tested material. Fusarium head blight is only tested in laboratory only on artificial medium and not on the plants e.g. wheat or barley. I cannot provide any comments on statistical analyses, you should seek the help of an statistician. As negative control they used only pure water in one stage in another stage authors mention methanol mixed with water.
Response: Thank you very much for your helpful suggestions and valuable input in our research manuscript. We also have carefully modified the whole manuscript. In our previous studies, we had implicated melatonin homolog, 5-methoxyindole, could directly induce the plant immune system to increase the disease resistance against plant pathogens as same as melatonin (Kong M., et al. 2021). However, the direct inhibition mechanism of melatonin homolog, 5-methoxyindole, has not been investigated in plant pathogens, which could be a potential compound to protect crop production from the stress of plant pathogens in the future. Then, we investigated the direct inhibition mechanism of melatonin-homolog 5-methoxyindole in our manuscript.

Round 2
Reviewer 2 Report
Dear Authors,
After revision, I accept the author's amendments and I ask for publication of this manuscript in its current form.
Author Response
Reviewer 2
After revision, I accept the author's amendments and I ask for publication of this manuscript in its current form.
Response: Thank you very much for your review of our manuscript and we are very appreciating to get you approve of our submitted manuscript.
Reviewer 3 Report
See minor corrections in abstract and in page 3 lin3 97.
Author Response
Reviewer 3
See minor corrections in abstract and in page 3 lin3 97.
Response: Thank you very much for your helpful suggestions and valuable input in our research manuscript. We have modified and corrected our whole manuscript, including the abstract and in page 3 lin3 97.